# OpenReview forum: "Analyzing and Improving Chain-of-Thought Monitorability Through Information Theory"
_ICLR.cc/2026/Conference — ICLR 2026 Conference Desk Rejected Submission_

### Official Review · Reviewer_pTo5 · 2025-10-31

**Soundness:** 3
**Presentation:** 4
**Contribution:** 3
**Rating:** 6
**Confidence:** 4

**Summary:**

- This paper provides an information-theoretic analysis of CoT monitoring, converting this analysis into a practical mitigation that can help improve monitorability during training
- The conditions in which CoT can provide a performance uplift are formalized. It is shown that strictly positive uplift in CoT monitorability requires additional positive mutual information between the CoT and the output, beyond what the prompt provides
- Next, the paper considers approximate CoT monitors, and states that the errors in an approximate monitor come from: (i) information gap - how well the monitor can predict the outputs of the model given the prompt and CoT; and (ii) elicitation error - how well the monitor can approximate the true output monitoring function.
- A solution that bears on 'information gap' is proposed, one that train the model to produce more informative CoTs. Specifically, an oracle-based reward, and a label-free reward term are porposed. The label free reward is based on an estimate of the conditional mutual information in the models CoT.
- Their mitigations are tested in an MBPP setup with a hackable task reward, while also training against a CoT monitor. Here, the main finding is that using the mitigation reward terms can prevent reward hacking in this setting.

**Strengths:**

- The contributed information-theoretic framework for CoT monitorability adds a novel and potentially useful and clarifying perspective to the field. Formalizing the importance of mutual information between CoT and outputs is useful for thinking about the problem. Similarly, concretely and formally identifying two distinct sources of CoT monitoring errors is useful for designing mitigations/improvements.
- The analyses have led to a novel and practical mitigation in the form of the proposed mutual information reward. In the provided experiments, this mitigation shows promising signs of life, greatly reducing MBPP reward hacking rates when training against a CoT monitor.
- The writing is very clear throughout.
- Overall, this theory and experiments could be significant and influential in driving future important research along similar directions. I am excited to see future work further investigating the efficacy of the mutual information reward, and other similar rewards that could improve the monitorability of the model's CoT.

**Weaknesses:**

**Limited experiments.**

The current set of experiments feel very limited in breadth and depth. This makes it hard for me to gauge how promising the MI reward truly is. I will outline additional experiments I would like to see below. Ideally, this paper would add a more thorough set of empirical results.

**Theory yields somewhat expected takeaways.**

While the provided theory is well executed and presented, the main takeaways from this theory are perhaps not overly surprising or insightful. Indeed Section 3.2 findings have been shown empirically by [Emmons et al (2025)](https://arxiv.org/abs/2507.05246).

The takeaways I am referring to are: (1)  "task requiring a CoT is a necessary, but not sufficient, condition for CoT
monitorability", and (2) the sources of error are information gap and elicitation error.

Nevertheless, I still maintain it is valuable to highlight and fomralize these, as done in the paper.

**Confusions regarding CoT being necessary for a task.**

In contribution 1, line 045, you state “...use it to show that a task requiring a CoT is a necessary, but not sufficient, condition for CoT monitorability”. Here, “a task requiring CoT” implies it is a task where the use of CoT is required to get good task performance, as in Emmons et al (2025). However, your analysis is based on the MI between CoT and output, and does not seem to consider the extent to which CoT is necessary for obtaining high task reward. It seems to me that the MI between CoT and output can still be high, even when CoT is not strictly necessary for the task? (i.e., even when the model can maximize task reward without the use of CoT). It seems important to make this distinction clear throughout the paper.

Likewise, it would be useful to emphasize whether the MBPP experiment setting is one where use of CoT is necessary to obtain high task reward.

**Unclear how previously empirically observed CoT monitoring failure modes relate to the provided theory.**

In general, the paper does not seem to consider CoT monitoring failure modes that have been highlighted in the literature. For example, [Arnav et al (2025)](https://arxiv.org/abs/2505.23575) show "CoT traces can also contain misleading rationalizations that deceive the monitor". I would be interested to hear from the authors how their theory accounts for empirical failure modes like this, and whether this deserves some more discussion in the paper.

**Questions:**

**Possible additional experiments.**

There are additional experiments that could clarify my understanding of the promise of the MI reward:

I would like to see more experiments that directly show the “monitor accuracy” metric being higher after training against R_MI, versus baselines. Currently, the experiments in Figure 2 doesn’t seem to directly do this. For example, you could consider training runs where there is no R_monitor, and the reward is just R_task + R_MI. This could help properly disentangle improvements in monitorability from the reward hack metric. It could also be useful to present somewhere monitor accuracies for hack and non-hack solutions seperately

In general, I would probably prefer to see some theory moved into the appendix and more depth of analysis of the experimental results in the main paper.

I would be particularly interested to know whether CoT is necessary for the Qwen models used on the current coding tasks. It would be useful to include results in an experimental setting where CoT is necessary - in these settings the MI reward could hurt task performance and have a higher "alignment tax"?

It could be nice to have experiments with reasoning models, though not crucial.

It could be nice to have experiments in a setting where the monitor looks at both the CoT and the outputs.

In general, results in more environments would be informative regarding the generality of your empirical findings. At least some discussion of the simple-equations environment results in the main body could be useful.

**Other.**

The transition from the identified sources of error in the theory to the practical mitigations used could perhaps be better justified. Corollary 3.14 identifies policy-as-monitor as a solution to information gap, however the paper instead proposes the MI reward. Additionally, there is not much discussion regarding why the paper decides to focus on the information gap error in its experiments, and not the elicitation error? I think what is done in the paper is mostly very reasonable, but could perhaps be better justified and tied back more precisely to the prior theory.

---

> ### Author Response · Authors · 2025-11-24
> **Authors' Response: Part I**
>
> We sincerely thank the reviewer for their helpful feedback and for noting “theory and experiments could be significant and influential in driving future important research along similar directions”. In the [global comment](https://openreview.net/forum?id=6whdepL0v1&noteId=vMundOwdOr), we have provided an overview of new experimental results added to the paper (including new environments and new models), and any notable revisions made to the paper.
>
> We provide our response to your review below.
>
> **Multiple new experiments have been added in the paper.**
> Thank you for your suggestions regarding additional experiments. In the revised copy of the paper. We have added the following experiments.
> - Experiments in an additional environment based on BigMATH dataset (Figure 2b).
> - Experiments with two additional models in MBPP, SmolLM3-3B and Deepseek-R1-Distill-LLaMa-8B (Figure 8). Note that R1-Distill-LLaMa is a reasoning model.
> - Experiments with 7B Qwen model in simple-equations (Figure 4).
> - Experiments that show monitor accuracy improvements from $R\_{MI}$ in the absence of monitor pressure (Figure 15 and Figure 2b).
> - Experiments that evaluate the ‘alignment tax’ of $R\_{MI}$ (Figures 14 and 22).
> - Extensive ablations of the hyperparameters a,b of the $R\_{MI}$ reward (Figures 9-12 and 21).
> - Experiments that evaluate the effect of $R\_{MI}$ on completion lengths (Figures 16-18).
>
> **Novel insights provided by the theory.**
> Thank you for appreciating the value our formalism provides. We agree that some results are not that surprising, but would respectfully like to point out that our theoretical framework does provide some novel insights:
>
> 1. **Policy-as-monitor optimality**: Understanding the optimal setup for CoT monitoring of steganographic models is an open problem, and was only explored empirically in the concurrent work of Stickland & Korbak recently. Our theoretical result in this regard (Corollary 3.13) is thus a well-timed contribution.
> 2. **Our theory is insightful about what attributes are monitorable:** Furthermore, our analysis provides insights about what kind of attributes are monitorable (last two paragraphs of section 4.2). This aspect has not been explored at all (empirically or via theory) in prior works.
> 3. **Mechanistic explanation of how CoT obfuscation arises:** Our theory provides a mechanistic explanation of CoT obfuscation results reported by Baker et al.
> 4. **Theoretical grounding for our training objectives:** Finally, we note that our theory provides justification for our proposed rewards: oracle reward indirectly maximizes $I(Y;Z|X)$ while $R\_{MI}$ is motivated as a practical proxy.
>
> **Text regarding link between CoT monitorability and CoT being necessary for a task has now been improved.**
> Thank you for catching this error. We have now fixed this in the updated paper, and clarified that the necessary condition for CoT monitorability is positive MI between O and Z, which is fulfilled in the cases where CoT is necessary for the task, but also in some other cases.
>
> **Evaluations regarding the necessity of CoT for MBPP/BigMATH?**
> In Appendix D (Table 2 & 3\), we have included results regarding whether CoT is necessary for Qwen models on MBPP and BigMATH datasets. While CoT is quite helpful for BigMATH, it is not necessary for MBPP based on these results. Notably, $R\_{MI}$ remains equally effective across both environments.

---

> > ### Author Response · Authors · 2025-11-24
> > **Authors' Response: Part II**
> >
> > **Discussion on relation between previously empirically observed CoT monitoring failure modes relate to the provided theory.**
> > Under our theory, a model can evade a CoT monitor through three mechanisms:
> >
> > 1. Removing the relevant (Shannon) information entirely from the CoT, i.e., making $I(Y;Z|X) \\rightarrow 0$.
> > 2. Increasing the information gap – i.e., reducing the usable information available to a bounded monitor.
> > 3. Increasing the elicitation error for the monitor – i.e., changing the CoT such that the (static) monitor’s ability to simulate the relevant monitoring function is disrupted.
> >
> > In practice, we are aware of three failure modes of CoT monitoring observed in practice:
> >
> > I. Information removal/obfuscation (Baker et al.).
> > II. Stegographic CoT (Stickland and Korbak, Emmons et al. and [Farquhar et al](https://arxiv.org/abs/2501.13011).).
> > III. Robustness failures like misleading rationalizations (Arnav et al.), or prompt-injections ([Terekhov et al.](https://arxiv.org/abs/2510.09462v1)).
> >
> > (1), (2) and (3) respectively map to the failure modes (I), (II) and (III). Of these, given the information theoretic focus of the paper, (I) and (II) feature prominently in the paper. **We have now added a sentence in Section 3.3 (page 5, lines 238-241) that gestures at the relation between elicitation error term and the robustness failures.** Though we hope that a future work investigates such failures more closely in the future.
> >
> > Regarding Arnav et al: The case of misleading rationalization shown in Figure 2 (right) of Arnav et al. is the classic case of an ML model focusing on a non-robust feature (misleading statement in CoT in this case) as opposed to focusing on more robust features present in the model’s output. Though we note that this is because in the case of Arnav et al., CoT monitors can observe both CoT and the output; if *only* CoT was available, then this failure mode would be due to the information gap.
> >
> > We note that Terekhov et al. do not strictly attack CoT monitors, but we use it as a reference as the high level takeaway regarding llm-as-monitor being vulnerable to typical adversarial attacks is similar.
> >
> > **Including more experimental analysis in the main paper.**
> > For the camera-ready (if the paper is accepted), the page limit is 10 pages. We will use the additional page to move experimental analysis from the appendix to the main paper as the reviewer suggested.
> >
> > **Better transition from identified sources of error in the theory to the practical mitigations.**
> > We have added a paragraph at the top of section 4 that we believe improves the transition from theory to the empirical work done in the paper.
> >
> > We hope that our response has adequately addressed your concerns. Given the improvements made to the paper, we hope that you would consider revising your score and confidence upwards and championing for the paper in any internal discussions reviewers and ACs might have.

---

### Official Review · Reviewer_m7dV · 2025-10-31

**Soundness:** 1
**Presentation:** 2
**Contribution:** 1
**Rating:** 0
**Confidence:** 4

**Summary:**

1) The paper’s entire theoretical framework makes a set of assumptions which, IIUC, **rule out the value of CoT monitoring a-priori**. In particular, sections 3.1-3.2 assume that:

A) Z (CoT) can only influence Y (attribute being monitored) via O (the model’s action).

B) The monitor is bayes-optimal.

But in this setting, it is *theoretically impossible* for the CoT to provide information to the monitor beyond that contained in the model’s action. Why bother monitoring the CoT at all, rather than the action?

Furthermore, the optimal CoT in terms of monitorability would be one which contains all of the information in O. The model could just *repeat O as its CoT*, and this would provide the maximum possible uplift as defined in Definition 3.7.

The paper attempts to improve the realism of its setup with subsection 3.3, which introduces “approximate CoT monitors”, accounting for monitors which may have limited capacity and computation. It then decomposes monitor error into two components: the “information gap” measuring how well the monitor’s internal model q(O | X,Z) matches the agent’s true policy, and elicitation error, measuring how well the monitor can approximate the true output monitoring function.

But in implicitly treating these terms as independent, the paper ends up in the same position as before: it is *theoretically impossible* for Z to provide information about Y to the monitor beyond that contained in O. Why monitor the CoT at all? The paper even seems to state this as a result: Corollary 3.14 shows that using the agent’s own policy to predict its own outputs achieves zero information gap. In other words, according to the paper’s assumptions, monitoring the agent’s actual actions is optimal; monitoring the CoT can’t possibly add value over just monitoring actions!

2) The paper attempts to draw connections between its results and existing empirical work in “Do CoTs that matter for the task also help monitoring?”, claiming its theoretical analysis shows “that the necessity of CoT reasoning for task completion is a necessary precondition for the monitorability of any attribute Y of the final output.” But proposition 3.11 only shows that positive uplift requires positive *mutual information* between O and Z, not that Z is necessary for O. In particular, consider the following scenario:

A) First, the LLM performs the task and decides on its output O internally in a single forward pass, without any necessary CoT.

B) Second, the LLM outputs O as its CoT.

C) Finally, the LLM outputs O as its final output.

The LLM’s CoT and its final output are perfectly correlated, and will therefore have the maximum possible mutual information. But the CoT had no necessity for the task whatsoever.

3) The empirical results, while interesting and suggestive of a trend, only evaluate two models on a single environment. Evaluating additional environments would provide more confidence in the robustness of the results.

4) The monitor for the RL experiments only observes the CoT. The experiments should also compare to a monitor which observed the actions, and a monitor which observes both actions and CoT, to show whether the CoT is providing value above simple action monitoring.

In particular, it seems like the current reward function would incentivize CoTs to just explicitly state the action that they’re going to take; but then why not just monitor actions?

**Strengths:**

1) The paper’s empirical results are interesting, and Figure 2 does a good job at communicating the value of the mutual information reward in the given environment.

2) The discussion of steganography in Appendix A, and the empirical identification of examples, is interesting (although it also highlights the limitations of the paper’s theoretical justifications for optimizing MI).

**Weaknesses:**

1) The paper’s entire theoretical framework makes a set of assumptions which, IIUC, **rule out the value of CoT monitoring a-priori**. In particular, sections 3.1-3.2 assume that:

A) Z (CoT) can only influence Y (attribute being monitored) via O (the model’s action).

B) The monitor is bayes-optimal.

But in this setting, it is *theoretically impossible* for the CoT to provide information to the monitor beyond that contained in the model’s action. Why bother monitoring the CoT at all, rather than the action?

Furthermore, the optimal CoT in terms of monitorability would be one which contains all of the information in O. The model could just *repeat O as its CoT*, and this would provide the maximum possible uplift as defined in Definition 3.7.

The paper attempts to improve the realism of its setup with subsection 3.3, which introduces “approximate CoT monitors”, accounting for monitors which may have limited capacity and computation. It then decomposes monitor error into two components: the “information gap” measuring how well the monitor’s internal model q(O | X,Z) matches the agent’s true policy, and elicitation error, measuring how well the monitor can approximate the true output monitoring function.

But in implicitly treating these terms as independent, the paper ends up in the same position as before: it is *theoretically impossible* for Z to provide information about Y to the monitor beyond that contained in O. Why monitor the CoT at all? The paper even seems to state this as a result: Corollary 3.14 shows that using the agent’s own policy to predict its own outputs achieves zero information gap. In other words, according to the paper’s assumptions, monitoring the agent’s actual actions is optimal; monitoring the CoT can’t possibly add value over just monitoring actions.

2) The paper attempts to draw connections between its results and existing empirical work in “Do CoTs that matter for the task also help monitoring?”, claiming its theoretical analysis shows “that the necessity of CoT reasoning for task completion is a necessary precondition for the monitorability of any attribute Y of the final output.” But proposition 3.11 only shows that positive uplift requires positive *mutual information* between O and Z, not that Z is *necessary* for O. In particular, consider the following scenario:

A) First, the LLM performs the task and decides on its output O internally in a single forward pass, without any necessary CoT.

B) Second, the LLM outputs O as its CoT.

C) Finally, the LLM outputs O as its final output.

The LLM’s CoT and its final output are perfectly correlated, and will therefore have the maximum possible mutual information. But the CoT had no necessity for the task whatsoever.

3) The empirical results, while interesting and suggestive of a trend, only evaluate two models on a single environment. Evaluating additional environments would provide more confidence in the robustness of the results.

4) The monitor for the RL experiments only observes the CoT. The experiments should also compare to a monitor which observed the actions, and a monitor which observes both actions and CoT, to show whether the CoT is providing value above simple action monitoring.

In particular, it seems like the current reward function would incentivize CoTs to just explicitly state the action that they’re going to take; but then why not just monitor actions?

**Questions:**

1) What are potential reasons one might want to monitor CoT, rather than just monitoring actions? Could you account for these theoretically?

---

> ### Author Response · Authors · 2025-11-24
> **Authors' Response: Part I**
>
> We thank the reviewer for their detailed feedback. However, **the primary critique in the review regarding the soundness of the theoretical framework contains several fundamental misunderstandings and multiple mischaracterizations of our results that we rebut below.**
>
> The reviewer’s first criticism is related to the two assumptions we make:
>
> 1. Z (CoT) can only influence Y (attribute being monitored) via O (the model’s action).
> 2. The monitor is bayes-optimal.
>
> Both these assumptions are highly reasonable at the object level, and indeed are not criticized at all on object level by the reviewer.
>
> Assumption 1 is justified on the basis that the goal of monitoring is typically to infer the properties of the output and is how the monitoring has been employed in prior works, e.g., [Baker et al.](https://arxiv.org/abs/2503.11926) and [Arnav et al.](https://arxiv.org/abs/2505.23575).
>
> Similarly, the assumption that the monitors are Bayes-optimal is made **because we want to understand the limits of CoT monitoring.** How useful CoT monitoring is remains an open, and highly contested, question in the community. Papers have been written both in favor of [usefulness of](https://arxiv.org/abs/2507.11473) CoT for monitoring and auditing and [against it](https://fbarez.github.io/assets/pdf/Cot\_Is\_Not\_Explainability.pdf) by highly respected set of authors.
>
> Our work makes progress on this question by examining the information-theoretic limits of CoT monitoring which necessitates the optimality assumptions made in the work. In particular, we show that when comparing an \\emph{oracle} CoT monitor to an \\emph{oracle} output monitor (under the assumption that the monitored attribute is an attribute of the output), the oracle CoT monitor can match the oracle output monitor, \\emph{whenever the CoT is sufficiently informative about the output for that attribute, i.e., $I(Y;O|X,Z)=0$}. This formal result aligns with intuitively motivated criteria like coverage in concurrent work (https://arxiv.org/abs/2510.23966).
> **On the whole, our results clarify the conditions for CoT monitorability to work and in no way should be taken to conclude that ‘CoT monitoring has no value’.** We would like to emphasize we are working with theoretical objects that have theoretical properties. The limits of CoT monitoring that we identify relate to the limits of CoT monitoring that are observed in practice, but in no way rule out the value of CoT monitoring. As we discuss below, **CoT monitoring can provide value DESPITE these limitations**.
>
> \> In other words, according to the paper’s assumptions, monitoring the agent’s actual actions is optimal; monitoring the CoT can’t possibly add value over just monitoring actions\!
>
> When the attribute being monitored is an attribute of the output, then the fact that CoT can not provide any *Shannon* information beyond the output is a mathematical fact. This will hold true in any setup where the monitored attribute is an attribute of the model output (which must follow CoT due to the autoregressive nature of the decoding in LLMs) as it generally is in prior empirical works (Baker et al., Arnav et al., Emmons et al.). **However, it is not necessary for CoT to provide additional *Shannon*** **information (over a model’s outputs) to be useful in practice.** In general, our theory, in no way, stipulates that **practical CoT monitors** can not outperform **practical output monitors.** We have added a new section 3.3.1 in the paper that provides an explanation for this based on the idea of *usable* information.
>
> To summarize that discussion, Shannon information is a theoretical construct that presumes observers (monitors in our case) have infinite capacity and compute. However, practical monitors have limited capacity and compute. Therefore, what matters in practice is *usable* information, i.e., how easily a constrained monitor can extract the relevant signal  (see [https://arxiv.org/abs/2002.10689](https://arxiv.org/abs/2002.10689) for an overview). That’s why a CoT can help even when it provably contains no additional *Shannon* information beyond the final output. The CoT can represent the same underlying signal in a way that may make it considerably easier for a practical monitor to read out.
>
> On the whole, we vehemently disagree with the assertion that our theoretical framework is not sound, or that the assumptions we make are unreasonable. The value of our theoretical framework has also been acknowledged and appreciated by other reviewers.
>
> - “The scope and direction of the paper is interesting and timely: understanding how to reliably monitor LLM reasoning is a critical, unsolved problem.” – Reviewer NPUK
> - “The contributed information-theoretic framework for CoT monitorability adds a novel and potentially useful and clarifying perspective to the field. Formalizing the importance of mutual information between CoT and outputs is useful for thinking about the problem.” – Reviewer pTo5

---

> ### Author Response · Authors · 2025-11-24
> **Authors' Response: Part II**
>
> > The paper claims its theoretical analysis shows “that the necessity of CoT reasoning for task completion is a necessary precondition for the monitorability of any attribute Y of the final output.” But proposition 3.11 only shows that positive uplift requires positive mutual information between O and Z, not that Z is necessary for O.
>
> We have modified the discussion around proposition 3.11 (now proposition 3.10 in the paper) and added a remark that addresses this critique.
>
> > The empirical results, while interesting and suggestive of a trend, only evaluate two models on a single environment. Evaluating additional environments would provide more confidence in the robustness of the results.
>
> We have added a number of new empirical results that provide further evidence for the efficacy of $R\_{MI}$, including a third environment and two more models. Please see the [global comment](https://openreview.net/forum?id=6whdepL0v1&noteId=vMundOwdOr) for an overview of all the new experiments now included in the paper.
>
> > The monitor for the RL experiments only observes the CoT. The experiments should also compare to a monitor which observed the actions, and a monitor which observes both actions and CoT, to show whether the CoT is providing value above simple action monitoring.
>
> **The goal of our paper is not to show the value of CoT monitoring relative to action monitoring.** This has already been done in the prior empirical works of Baker et al. and Arnav et al. The goal of our work is to improve the understanding of the limits and the failure modes of CoT monitoring. Our experiments show that CoT obfuscation observed by Baker et al. is a consequence of actions and CoT becoming decorrelated and that using $R\_{MI}$ can prevent such obfuscation from arising.
>
> > It seems like the current reward function would incentivize CoTs to just explicitly state the action that they’re going to take; but then why not just monitor actions?
>
> If we solely optimize for $R\_{MI}$, then the optimal thing for LLM indeed would be to duplicate the output as CoT. However, there are opposing pressures from the task reward, format rewards and prior pretraining **such that we do not observe this failure mode in our experiments.**
>
> Regarding, why not just monitor actions, see the response below.
>
> > Question: What are potential reasons one might want to monitor CoT, rather than just monitoring actions? Could you account for these theoretically?
>
> Understanding what monitoring scheme is appropriate for what setting is an active area of research. However, in our opinion, there are at least three reasons why one might want to monitor CoT over (and/or in addition to monitoring) the actions when the properties we care about are properties of the action:
>
> 1. Representational advantage of CoT: Actions/outputs can be highly complex and difficult to interpret in many scenarios. In contrast, in CoT, the same information is generally available in plain English which can make the monitoring task more feasible. In general, if there is a task for which output monitors have higher misspecification error than the corresponding CoT monitor, that could make the case for using CoT monitoring for the task as opposed to output monitoring.
> 2. Dynamic monitoring pipelines: CoT is produced before the final action. Monitoring CoT can therefore let you intervene early (e.g., block or escalate to output monitor) if you suspect the output to be undesirable. Getting a predictive signal earlier is generally preferable in most real world monitoring tasks.
> 3. Adversarial obfuscation at the output level: Generally, LLMs are trained using outcome based rewards, this can result in LLMs obfuscating their outputs to avoid output monitors. CoT in such cases may still provide a reliable monitoring signal.
>
> We now provide a theoretical account of 1 in Section 3.3.1 and Appendix F.1. 2 is a standard engineering principle and the theoretical accounting of 3 is out of the scope of the current work.
>
> **Given our comprehensive rebuttal, we hope that all your concerns, especially the ones regarding the soundness of our theoretical framework and the value of CoT monitoring, have now been addressed. If so, we kindly ask you to please revise your score. If not, we would appreciate clear communication from you regarding what aspects of the work are not sound.**

---

> ### Comment · Reviewer_m7dV · 2025-11-27
>
> Thanks for your reply.
>
> > Both these assumptions are highly reasonable at the object level, and indeed are not criticized at all on object level by the reviewer.
>
> Apologies if I didn’t make this clear - the problem is the bayes-optimality assumption. Consider the reasons you gave for monitoring CoT:
>
> 1. Representational advantage of CoT
> 2. Dynamic monitoring pipelines
> 3. Adversarial obfuscation at the output level
>
> 1 and 3 completely disappear in the context of bayes-optimality: a bayes-optimal monitor will be entirely insensitive to differences in representation, and to obfuscation. Why would you make an assumption which nullifies the primary motivating value of the approach you’re considering?
>
> Indeed, the two assumptions in conjunction imply that *no white-box interpretability technique can possibly add information value over black-box action monitoring*. Why study an interpretability technique in this regime?
>
> > When the attribute being monitored is an attribute of the output, then the fact that CoT can not provide any Shannon information beyond the output is a mathematical fact.
>
> Yes, this is a trivial property of the causal graph.
>
> > the assumption that the monitors are Bayes-optimal is made because we want to understand the limits of CoT monitoring
>
> But this theory doesn’t imply much about the limits of CoT monitoring, because…
>
> > it is not necessary for CoT to provide additional Shannon information (over a model’s outputs) to be useful in practice.
>
> Right! This is my central concern: the paper spends 6 pages deriving results which are 1) mathematically trivial, and 2) hardly imply anything for real CoT monitoring.
>
> > To summarize that discussion, Shannon information is a theoretical construct that presumes observers (monitors in our case) have infinite capacity and compute. However, practical monitors have limited capacity and compute. Therefore, what matters in practice is usable information, i.e., how easily a constrained monitor can extract the relevant signal (see https://arxiv.org/abs/2002.10689 for an overview). That’s why a CoT can help even when it provably contains no additional Shannon information beyond the final output. The CoT can represent the same underlying signal in a way that may make it considerably easier for a practical monitor to read out.
>
> Agree. I can definitely imagine a very interesting version of this paper, which starts by briefly stating the bayes-optimality results (with derivations in appendix) and then spends the bulk of its time on constructing a theoretical model for “usable” information, which is what’s actually relevant for practical monitoring.
>
> > we vehemently disagree with the assertion that our theoretical framework is not sound
>
> I think the bayes-optimality framework is sound, just not practically relevant.
>
> I’m still concerned about Section 3.3 - the claim that “a model trained to produce steganographic CoTs is best monitored by itself” assumes that elicitation error and information gap are independent, and so minimizing information gap also minimizes misspecification error. But of course, the point of CoT (and any interpretability technique) is that it should *reduce elicitation error* compared to action monitoring, and this reduction in elicitation error may overcome the information gap incurred.
>
> > or that the assumptions we make are unreasonable
>
> Right, this is the crux - I think bayes-optimality is unreasonable when your goal is to study an interpretability technique.
>
> > We have modified the discussion around proposition 3.11
>
> Thanks, the new discussion looks like a more accurate description of the result (though the practical connection is correspondingly weaker).
>
> > We have added a number of new empirical results
>
> Thanks, the additional environment and models definitely improve the robustness of the findings. I do think that the empirical results are the strongest part of the paper, and I wish more of the main paper had focused on these findings.

---

> > ### Author Response · Authors · 2025-12-02
> > **Authors' Further Response: Part 1**
> >
> > ## Claims about section 3.3 not being sound are false
> >
> > > I’m still concerned about Section 3.3 \- the claim that “a model trained to produce steganographic CoTs is best monitored by itself” assumes that elicitation error and information gap are independent, and so minimizing information gap also minimizes misspecification error.
> >
> > The quoted claim that “a model trained to produce steganographic CoTs is best monitored by itself” refers to the empirical findings of Stickland & Korbak; quoting from their [blogpost](https://www.lesswrong.com/posts/ZEdP6rYirxPxRSfTb/if-you-can-generate-obfuscated-chain-of-thought-can-you), “We find that, in most cases, the finetuned models \[which are generating steganographic reasoning\] are better monitors than the models they are finetuned from”.
> >
> > The relevant claim that we make regarding our theory is “if and when, an LLM may be using steganographic reasoning, information gap can become the dominant term. In such a case, it may be advisable to use the policy model as monitor model as well (a setup we call policy-as-monitor). This setup has the advantage that it eliminates the information gap between the monitor and the agent.” and Corollary 3.14 clearly says that elicitation gap remains in this case. This gap can increase/decrease relative to some other monitor if $\\pi$ is relatively worse/better at approximating $v$. **Thus, we nowhere assume or claim that “elicitation error and information gap are independent, and so minimizing information gap also minimizes misspecification error”.**
> >
> > > But of course, the point of CoT (and any interpretability technique) is that it should *reduce elicitation error* compared to action monitoring, and this reduction in elicitation error may overcome the information gap incurred.
> >
> > While this intuition seems reasonable, it is not at all guaranteed that CoT should reduce elicitation error. See Section 3.1 in [Arnav et al.](https://arxiv.org/pdf/2505.23575) that gives examples of cases where monitoring CoT **can increase the elicitation error** relative to output/action monitoring.
> >
> >
> > ## Practical relevance and utility of Bayes-optimality framework (section 3.2)
> >
> > > I think the bayes-optimality framework is sound, just not practically relevant.
> >
> > The Bayes-optimality framework is useful for characterizing the limits of CoT monitoring with respect to this optimal setting. Analyzing an optimal setting is a standard approach for understanding the limits of some procedure. While the reviewer claims that section 3.2 is not practically relevant, we use it to provide an explanation of how CoT obfuscation arises (a phenomenon observed in Baker et al.). The results in this section also provide theoretical grounding for our training objectives, which, the reviewer agrees, have clear practical relevance.
> >
> > **In practice, adopting the Bayes-optimality framework enables us to derive the other results in the paper, including those that characterize the gap with practical monitors and those that lead to our practical training objective.**

---

> > > ### Author Response · Authors · 2025-12-02
> > > **Authors' Further Response: Part 2**
> > >
> > > ## The paper is already structured very similarly to the reviewer’s suggestion
> > >
> > > > This is my central concern: the paper spends 6 pages deriving results which are 1\) mathematically trivial, and 2\) hardly imply anything for real CoT monitoring.
> > > > I can definitely imagine a very interesting version of this paper, which starts by briefly stating the bayes-optimality results (with derivations in appendix) and then spends the bulk of its time on constructing a theoretical model for “usable” information, which is what’s actually relevant for practical monitoring.
> > >
> > > This is basically the current structure. The results regarding Bayes-optimality (section 3.2) only span \<1 page (with derivations already moved to the appendix) and rest of the time is spent on:
> > >
> > > (i) characterizing the gaps between Bayes-optimal CoT monitors and practical CoT monitors (which leads to insights about sources of error for practical monitors, and an explanation of the empirical observations of Stickland & Korbak).
> > > (ii) Using the usable information framework to characterize conditions under which a practical CoT monitor may outperform practical output monitors.
> > > (iii) Proposing reward functions for improving CoT monitorability.
> > >
> > > We then provide extensive experimental validation of our proposed reward functions in the rest of the paper and the Appendix. The reviewer notes that these results are “interesting” and robust.
> > >
> > > **The reviewer’s claims reduce the theoretical contribution of the paper to a single section (section 3.2), disregarding the rest.**
> > >
> > > **Moreover, we vehemently disagree that our results have no relevance for real CoT monitoring.** As we explain in our reply to reviewer NPUK, our theory provides at least the following novel insights:
> > >
> > > 1. **Decomposition of monitor error**: Proposition 3.13 in Section 3.3 provides decomposition of practical monitor error into two distinct sources of errors.
> > > 2. **Policy-as-monitor**: Our theoretical result in Corollary 3.14 is a well-timed contribution for understanding the optimal setup for CoT monitoring of steganographic models, which is an open problem only recently explored in Stickland & Korbak.
> > > 3. **Monitorably attributes:** Our analysis provides insights about what kind of attributes are monitorable (last two paragraphs of section 4.2).
> > > 4. **Mechanistic explanation of how CoT obfuscation arises:** Our theory provides an explanation of the CoT obfuscation results reported by Baker et al., through the non-sufficiency condition of proposition 3.10.
> > > 5. **Theoretical grounding for our training objectives:** Our theory provides justification for our proposed rewards: oracle reward directly maximizes $I(Y;Z|X)$ while $R\_{MI}$ is a practical proxy.
> > >
> > > ## The review’s content is not consistent with the extreme scores
> > >
> > > The paper is given a soundness score of 1 (poor) and an overall recommendation of 0 (strong reject) by the reviewer. However:
> > >
> > > * No concrete error has been identified in any theorem, proof, or empirical methodology.
> > > * In the follow-up, the reviewer explicitly acknowledges that the Bayes-optimal framework is sound (which was the main focus of their critique in their initial review).
> > >
> > > **Given that the empirical results are described as “interesting” and robust, and that the remaining disagreement is about *emphasis* and *relevance* rather than correctness, these extreme scores are difficult to reconcile with the content of the review.**
> > >
> > > ## The other reviewers’ critiques were minor and have been addressed
> > >
> > > Finally, we note that the other two reviewers view the paper in a highly positive light (see their noted strengths). Their critiques were generally minor and have been addressed through the comprehensive additional results we have added to the paper during the rebuttal period.
> > >
> > > **We hope these clarifications show that  (i) our framework is technically sound, (ii) Bayes-optimal analysis plays a limited but principled role, and (iii) the main contribution lies in understanding and improving practical CoT monitoring.**

---

### Official Review · Reviewer_NPUK · 2025-11-03

**Soundness:** 3
**Presentation:** 2
**Contribution:** 2
**Rating:** 6
**Confidence:** 2

**Summary:**

This paper analyzes the “monitorability” of chain-of-thought (CoT) reasoning (i.e., how well we can predict if attributes of interest $Y$ like hacking/malicious intent/bias are present/factors in a model’s response $O$ to an input $X$ by examining the LLM’s reasoning trace $Z$). The main contribution of this paper is to ground this analysis in an information theoretic framework, and uses this grounding to make the (intuitive) argument that monitoring is possible only when the CoT $Z$ carries information that is relevant to the attribute $Y$, and that tasks requiring an informative CoT (high $I(O;Z|X)$) provide a necessary (but not sufficient) condition for this monitorability. From an applied perspective, the authors propose a label-free auxiliary reward that maximizes the mutual information between the output and the CoT to encourage models to produce informative CoTs (and thereby, improve the necessary but not sufficient condition for monitorability).

**Strengths:**

- The scope and direction of the paper is interesting and timely: understanding how to reliably monitor LLM reasoning is a critical, unsolved problem.
- The analysis leads to a practical label-free reward that appears to lead to modest improvements in monitorability (+ resistance to hacking when paired with monitoring reward + task reward).
- A precise language for discussing monitorability (and formalizing its necessary conditions) can be valuable for the community (however, the setup is arguably a bit long-winded in getting to the main conclusions — and not much space is left for the experiments).

**Weaknesses:**

- The theoretical argument is nicely done, but it’s unclear how much new insight it offers (e.g., it is intuitively obvious that a CoT must contain information relevant to the output + attribute of interest to be useful for monitoring).
- The paper proves that maximizing I(O, Z | X) is not sufficient, but this is not reflected in the proposed reward. What stops the model from gaming this reward by simply outputting the final answer in the thought?
- It’s not that obvious how well the MI reward would generalize to other tasks — the abstract and introduction motivate broad concepts like “harmfulness” and “bias” yet the experiments focus on fairly specific forms of hacking.  It’s unclear if maximizing I(O, Z | X) would be effective for more complex attributes (and if a model cannot learn to easily obfuscate the parts relevant to Y but keep parts relevant to O).

**Questions:**

- Could the authors elaborate on what prevents a model from maximizing $R_{\text{MI}}$ by producing a CoT $Z$ that is simply a copy of the output $O$? Is the <think>/<code> XML structure reward (mentioned briefly in Sec 5.1) required to prevent this?
- The MI estimator seems to be a critical, non-trivial component. How sensitive is the model to the choice of a and b? How necessary is it to normalizing by length?

---

> ### Author Response · Authors · 2025-11-24
> **Authors' Response**
>
> We thank the reviewer for their thoughtful and constructive feedback, and for noting that the “the paper is interesting and timely.” In the [global comment](https://openreview.net/forum?id=6whdepL0v1&noteId=vMundOwdOr), we have provided an overview of new experimental results added to the paper (including new environments and new models), and any notable revisions made to the paper.
>
> We provide a response to your review below.
>
> ### **Response to Weakness 1: Novel Theoretical Insights Provided by Theory**
>
> We acknowledge that the high-level intuition ("CoT must contain relevant information") is obvious. However, we believe that formal results that verify intuitions are nevertheless valuable and would also note that our theory does provide novel insights:
>
> 1. **Decomposition of monitor error**: Proposition 3.13 in Section 3.3 provides decomposition of monitor error into two distinct sources of errors. We do not know of any prior works that clearly makes a distinction between these two error sources.
> 2. **Policy-as-monitor optimality**: Understanding the optimal setup for CoT monitoring of steganographic models is an open problem, and was only explored empirically in the concurrent work of Stickland & Korbak recently. Our theoretical result in this regard (Corollary 3.14) is also a well-timed contribution.
> 3. **Our theory is insightful about what attributes are monitorable:** Furthermore, our analysis provides insights about what kind of attributes are monitorable (last two paragraphs of section 4.2).
> 4. **Mechanistic explanation of how CoT obfuscation arises:** Our theory provides a mechanistic explanation of CoT obfuscation results reported by Baker et al.
> 5. **Theoretical grounding for our training objectives:** Finally, we note that our theory provides justification for our proposed rewards: oracle reward directly maximizes $I(Y;Z|X)$ while $R\_{MI}$ is motivated as a practical proxy.
>
> ### **Response to Weakness 2: Explanation of why $R\_{MI}$ is not gamed**
>
> We propose two training rewards:
>
> 1. $R\_{Oracle}$ which directly maximizes for $I(Y;Z|X)$, and thus is guaranteed for improving the monitorability of the attribute Y.
> 2. $R\_{MI}$ is proposed as a practical approximation to $R\_{Oracle}$, and thus lacks any guarantees. However, it nevertheless results in improvement in the CoT quality vis-à-vis monitoring. Regarding why $R\_{MI}$ is not gamed; there are two reasons for this:
> 1. While duplicating the output as CoT might be optimal for $R\_{MI}$, it is unlikely to be optimal for $R\_{Task}$, especially, if CoT is necessary for performing well on the task, and thus is not optimal for $R\_{Total} \= R\_{MI} \+ R\_{Task}$.
> 2. Formatting reward enforces the model to produce CoT first, and then the output. This means that while generating CoT, the model does not *yet* know what the output is going to be and thus can not duplicate it. Other pretraining/posttraining processes applied to the LLM may also bias the LLM against producing the output directly inside \<think\>\</think\> tags.
>
> ### **Response to Weakness 3: Generalization to Complex Attributes**
>
> Section 4.2 includes some discussion in this regard – $R\_{MI}$ should work well for coarse-grained attributes, but may have limited access for more fine-grained attributes. Nevertheless, we agree that empirical evidence regarding the generalization of $R\_{MI}$ to complex attributes is lacking. We have revised both the abstract and introduction to remove the mention of harmfulness and bias. Though we note that detecting test-hacking like behaviors is an equally important and practical problem as well.
>
> ### Response to Questions
>
> **Why model does not copy output O into CoT**
> See the response to weakness 2 above.
>
> **Sensitivity to the choice of hyperparameters in R\_{MI}**
> We have now included extensive ablations over a, b parameters of $R\_{MI}$ in Figures 9-12. $R\_{MI}$ is generally effective across a broad range of a,b values, though extreme values can have undesirable effects (e.g., for Qwen-3B, low value of b does not reduce hacking). Please see C.1.2 for further discussion.
>
> **Normalizing R\_{MI} by length**
> Normalization by length is motivated by the fact that without it an LLM can gain ‘free’ MI by just increasing the length of the output (discussed in the paper in the section 4.2.1). Another issue is that because length can vary across samples and during training, with unnormalized MI, it makes the choice of optimal a,b values trickier.
>
> We once again thank you for your review, and hope that in light of our response and additional experimental results now included in the paper, you will consider revising your score and confidence. Please let us know if you have any further questions.

---

### Author Response · Authors · 2025-11-24
**Author's Response: Details on new empirical results and revisions to the paper**

In this general comment, we want to highlight important revisions made to the paper (**all major changes are highlighted using blue-colored text in the uploaded pdf**).

1. The discussion around Proposition 3.10 in “Do CoTs that matter for the task also help monitoring?” previously made a misleading claim. This has now been fixed.
2. In section 3.3.1 (and related Appendix F.1), we have now included a discussion on how practical CoT monitors could do better than practical output monitors.
3. In section 5 (and Figure 2), results on a new environment based on BigMath have now been added. Note that results for Qwen-3B on MBPP have now been moved to the appendix.

We have also added a number of new experiments that we provide a quick overview of below.

1. **Additional environment**. We have added experiments on an environment inspired by the BigMath-RL-Verified dataset. The goal is for the model to solve (difficult) math problems, in a setting where the environment contains a hack: negative valued answers are passed by the verifier. These experiments can now be found in the main paper (Figure 2b); results match those on the MBBP environment, highlighting the robustness of our method across environments.
2. **Additional models in the MBPP environment.** We have added results on SmolLM3-3B and DeepSeek-R1-Distill-Llama-8B to the Appendix B.1.1. These models follow broadly the same patterns as observed in the Qwen experiments: strong hacking tendencies under monitor pressure, which are mitigated by our MI reward term. **This highlights the robustness of our method across models.**
3. **Extended ablations of MI parameters** a and b on all models for MBPP and BigMath. These experiments show the sensitivity of our method to changes in hyperparameter values. On MBPP, we also present experiments that characterise the effect of these parameters on completion length.
4. **$R\_{Task} \+ R\_{MI}$ experiments** for the MBPP environment in Appendix B.3. These aim to answer reviewer pTo5’s question “does R\_mi improve monitorability in the absence of monitor pressure?”, confirming that well-tuned $R\_{MI}$ improves monitorability in this setting.
5. **$R\_{True} \+ R\_{MI}$ experiments** for the MBPP and BigMath environments in Appendix B.2 and Appendix C.2. These aim to characterise the \`alignment tax’ of our method: does optimizing R\_mi together with the true task reduce model performance compared to optimizing for the true task only. We observe some performance decay, most of which can be mitigated by proper tuning of MI parameters. We also include results in the Appendix that suggest that CoT is required for the Qwen 7B model in the BigMath environment, further strengthening our claim that $R_\text{MI}$ does not lead to significant CoT degradation here.

---

### Author Response · Authors · 2025-12-02
**Note to AC: The review by reviewer m7dV is largely unsound and not consistent with the extreme score**

We would like to note that the paper received an extreme score of 0 (with soundness score of 1) from reviewer m7dV. We do not think the score is justified and we argue that the critique in the review is largely unsound. Our detailed response to reviewer m7dV (in the corresponding author–reviewer thread) explains this carefully, in particular, our latest two comments [here](https://openreview.net/forum?id=6whdepL0v1&noteId=deJC9UgBto) and [here](https://openreview.net/forum?id=6whdepL0v1&noteId=SCiqUXQ8AZ). We emphasize that the content of the review is not consistent with the extreme score.

Given this, and in light of the substantially more positive assessments from the other reviewers, we would kindly ask the AC to carefully consider the soundness of this particular review.

---

### Note · Program_Chairs · 2026-01-17
**Submission Desk Rejected by Program Chairs**

The following references in this submission do not refer to real documents and/or have major errors in bibliographic information:

 Sumanth Tanneru and Srijan Kumar. On the limits of eliciting faithful chain-of-thought reasoning with in-context learning and fine-tuning. arXiv preprint arXiv:2404.14534, 2024.